

# Spatio-temporal variability of lightning activity in Europe and the relation to the North Atlantic Oscillation teleconnection pattern

David Piper[1,2] and Michael Kunz[1,2]

[1]Institute of Meteorology and Climate Research (IMK), Karlsruhe Institute of Technology (KIT), Karlsruhe, Germany
[2]Center for Disaster Management and Risk Reduction Technology (CEDIM), Karlsruhe, Germany

*Correspondence to:* Michael Kunz
(michael.kunz@kit.edu)

**Abstract.** Comprehensive lightning statistics are presented for a large, contiguous domain covering several European countries such as France, Germany, Austria, or Switzerland. Spatio-temporal variability of convective activity is investigated based on a 14-year time series (2001–2014) of lightning data. Based on the binary variable thunderstorm day, the mean spatial patterns of lightning activity and regional peculiarities regarding seasonality are discussed. Diurnal cycles are compared among several regions and evaluated with respect to major seasonal changes. Further analyzes are performed regarding interannual variability and the impact of teleconnection patterns on convection.

Mean convective activity across central Europe is characterized by a strong northwest-to-southeast gradient with pronounced secondary features superimposed. The zone of maximum values of thunderstorm days propagates southwestward along the southern Alpine range from April to July. Diurnal cycles vary substantially both between different months and regions, particularly regarding the incidence of nighttime lightning. The North Atlantic Oscillation (NAO) is shown to have a significant impact on convective activity in several regions, pointing to a crucial role of large-scale flow in steering spatio-temporal patterns of convective activity.

## 1 Introduction

Among the damage caused by lightning strikes, convection-related weather phenomena such as strong wind gusts, heavy rain, hail, or tornadoes often lead to major economic losses and pose a significant threat to human life (e.g. Kunz and Puskeiler, 2010; Peyraud, 2013; Piper et al., 2016). In several European countries or regions such as Switzerland or southern Germany, the largest share of losses by natural hazards is related to severe convective storms.

Due to their local-scale nature, convective storms and related phenomena are not entirely and homogeneously recorded over larger areas. For this reason, several studies have tried to establish a connection between convective events and different indirect climate data, so-called proxies. The temporal and spatial variability of convection is then studied using such an appropriately defined proxy. Several papers focusing on ambient conditions favorable for the formation of thunderstorms found a strong relation between several convective parameters and thunderstorm probability, especially for severe storms (e.g. Van Delden, 2001; Brooks et al., 2003; Kunz, 2007; Mohr and Kunz, 2013). More recent studies consider data from radar or satellite as a proxy for hail (e.g. Cecil and Blankenship, 2012; Nisi et al., 2016; Puskeiler et al., 2016; Junghänel et al., 2016). These studies



found a strong spatial variability of hail probability that is mainly governed by the distance to the ocean and by orographic flow deviations.

In our paper, we used lightning flashes as a proxy for convective activity, since the data are available over several years and the recordings exhibit a high level of homogeneity. Whereas first lightning climatologies were deduced from SYNOP records (e.g. Wakonigg, 1978; Cehak, 1980; Pelz, 1984; Cacciamani et al., 1995), the recently developed electromagnetic sensor networks allow for spatially homogeneous analyzes, such as the early study performed by Finke and Hauf (1996) with respect to Southern Germany. Several papers focusing on different European regions found various distinct regional and local structures of lightning probability or density (e.g. Schulz et al., 2005; Coquillat et al., 2013; Wapler, 2013; Czernecki et al., 2016). Besides spatial features, also the temporal variability on diurnal and annual timescales shows several peculiarities across the respective investigation areas (e.g. Soriano et al., 2005; Antonescu and Burcea, 2010; Gladich et al., 2011; Santos et al., 2013). Europe-wide investigations of lightning climatology (e.g. Holt et al., 2001; Anderson and Klugmann, 2014) have also been performed using very low frequency (VLF) networks implying a lower spatial detection accuracy. Satellite-based techniques allowed for developing global lightning climatologies (e.g. Christian et al., 2003; Beirle et al., 2014), but again at the expense of a reduced spatial resolution.

All studies cited above, however, employed relatively short time series, in most cases not exceeding a length of 6 years. In addition, they either are restricted to one country or even smaller domains, or exclude the analysis of small-scale features due to the lower resolution. Our paper is based on 14 years (2001–2014) of high-resolution low frequency (LF) lightning data for the summer half-year (April–September) covering a large contiguous area consisting of several European countries such as France, Germany, Austria, or Switzerland. This allows for comprehensive and reliable statistical analyses of convective activity under various geographical conditions including the influence of complex orography. In particular, the large sample facilitates the investigation of variability on interannual timescales. Furthermore, many studies employed the quantity lightning density as a measure for convective activity, which suffers from being sensitive to single severe events or outliers. This issue can be resolved by using the dichotomous variable thunderstorm day (TD). Its classical definition as a day with audible thunder or visible lightning at a station (WMO, 2001) has the major drawback of the detection range being limited and highly variable (e.g., Haklander and van Delden, 2003; Bissolli et al., 2007; Novák and Kyznarová, 2011). Based on lightning data, diverse subjective thresholds were used (e.g. Wapler, 2013). We introduce a new, objectified TD definition robust in the case of single severe events and simultaneously allowing for filtering days with sporadic, weak thunderstorms, which are outside the scope of this study.

The objectives of this paper are to develop a comprehensive, high-resolution lightning climatology for large parts of western Europe, and to thoroughly investigate the joint characteristics of both spatial and temporal modes of variability. This involves for instance discussing regional peculiarities regarding the annual and diurnal cycles of convective activity and addressing the seasonal dependence of diurnal lightning peaks. Moreover, we study several aspects of interannual variability such as spatial correlations of local multi-year TD time series and the impact of the North Atlantic Oscillation teleconnection pattern on convective activity.



The paper is structured as follows: In Section 2, we briefly describe the different data sets that were used and the methods applied. Section 3 presents our major results and discusses the most relevant drivers and underlying physical mechanisms that plausibly explain the spatio-temporal variability observed. In Section 4, we provide a short summary of the key points and

draw some conclusions from the results.

## 2   Data and methods

### 2.1   Model and observational data

Convective activity is investigated in a domain comprising the countries of Germany, Austria, Switzerland, the Netherlands, Belgium, Luxembourg, and France (Fig. 1). In addition, data are available for parts of neighboring countries such as the north-

westernmost area of Italy, the Spanish part of the Pyrenees, or the Bohemian Forest. The complex terrain of the investigation area including large parts of the Alps and several low mountain ranges such as the Black Forest and Swabian Jura (s), the Ore Mountains (x) and the Massif Central (e) allows to study lightning activity in the presence of strong altitudinal gradients, glaciated areas and complex, contorted deep valley systems. On the other hand, the North German Plain (v) exhibits only iso-lated low hills at altitudes little above sea level. Maritime influence can be expected along the coasts of North Sea and Atlantic

Ocean as well as the French part of the Mediterranean.

The investigations cover a 14-year period from 2001 to 2014. Since convective storms in Europe occur most frequently during the warm summer months, we restrict our analysis to the summer half-year (SHY) from April to September.

### 2.1.1   Lightning data

The spatial and temporal variability of convective activity is examined using data from the ground-based low frequency (LF)

lightning detection system BLIDS (BLitz-Informations-Dienst Siemens) being part of the European EUCLID (EUropean Co-operation for LIghtning Detection) network. The records have a temporal and spatial resolution of 1 ms and 1 km, respectively (Drüe et al., 2007). Information about both polarity and current strength is neglected, and flashes rather than strokes are studied. Since the LF operational range implies a significantly lower detection efficiency of cloud-to-cloud (IC) lightning (Pohjola and Mäkelä, 2013), only cloud-to-ground (CG) flashes are taken into account.

### 2.1.2   North Atlantic Oscillation index

Convective conditions and related thunderstorm activity across Europe tend to form larger-scale patterns that exhibit a large annual and interannual variability (e.g., Kunz et al., 2009; Bedka, 2011; Mohr et al., 2015). To examine whether this variability may be partly controlled by major large-scale teleconnection patterns, we considered monthly values of the North Atlantic Oscillation (NAO) index. Those data were provided by the U.S. National Oceanic and Atmospheric Administration (NOAA).

The calculation of the index values is based on rotated S-mode principle component analysis (PCA; Richman, 1986) applied to monthly mean standardized 500 hPa height anomalies (Barnston and Livezey, 1987; Hurrell and Deser, 2010) obtained from





the National Centers for Environmental Prediction – National Center for Atmospheric Research reanalysis (NCEP/NCAR1; Kalnay et al., 1996).

## 2.2 Statistical methods

### 2.2.1 Binary measure of convective activity: Thunderstorm day

Lightning density usually is defined as the mean daily flash total within a certain grid box. This quantity has been used by several studies to estimate thunderstorm activity (e.g., Schulz et al., 2005; Manoochehrnia et al., 2007; Antonescu and Burcea, 2010; Santos et al., 2012). However, the explanatory value of lightning density suffers from sometimes being dominated by single severe convective storms producing several 10,000 flashes. Sporadic severe storm days may result in potentially misleading conclusions about the spatial patterns of convective activity.

This problem can be circumvented by defining a dichotomous variable *thunderstorm day* (TD hereafter), which takes the value one, if the number of daily flashes within a grid box exceeds a given threshold. Filtering those days with a low number of flashes enables us to focus on days with more intense thunderstorms and neglecting weak convective events such as, for example, embedded convection (Fuhrer and Schär, 2005). The optimum threshold in our TD definition is determined by an objective method. For this, the domain is subdivided into a grid consisting of $10 \times 10$ km$^2$ equidistant cells. Excluding all days without any flash in a respective grid cell, the empirical probability density distribution is computed for each cell separately. Averaging over the entire domain yields the distribution shown in Fig. 2. The threshold is determined by choosing the integer value just above the interval of strongest curvature, yielding a lower threshold of 5 flashes per day within a grid cell. This threshold excludes the large number of weak events, but simultaneously yields sufficiently large sample sizes. Therefore, for all subsequent investigations a day (from 00 to 00 UTC on the next day) is classified as TD if at least 5 flashes were registered within a $10 \times 10$ km$^2$ grid cell.

### 2.2.2 Probability density function

To provide a rough overview of the diurnal lightning incidence in different subregions, 24 h flash totals are fitted to a theoretical probability density function. For this purpose, it is reasonable to consider all days with lightning events instead of setting a certain threshold. Due to high skewness expected in connection with a large number of days with only a small number of flashes, the two-parameter gamma distribution is an appropriate choice:

$$f(x) = \frac{\left(\frac{x}{k}\right)^{\alpha-1} \exp\left(-\frac{x}{k}\right)}{k\Gamma(\alpha)} \tag{1}$$

where $\Gamma$ is the gamma function and $\alpha$ and $k$ are shape and scale parameter, respectively (Wilks, 1995). The fitting procedure is performed using the method of L-moments, which is a more robust alternative to the conventional method of moments (Hosking, 1990). Goodness of fit is assessed by analyzing a quantile-quantile-plot (Wilks, 1995).





### 2.2.3 Dispersion and correlation of annual TD numbers

To determine the sample dispersion of the annual TD time series within a specific grid cell, we considered the coefficient of variation $\hat{v}$, which is defined by the sample standard deviation $\hat{\sigma}$ normalized by the sample mean $\hat{\mu}$, thus independent of the latter quantity (Kohn, 2006):

$$\hat{v} = \frac{\hat{\sigma}}{\hat{\mu}} \qquad (2)$$

Apparently, $\hat{v}$ exhibits a singularity for $\hat{\mu} \to 0$, which must be kept in mind when analyzing the dispersion in areas with weak convective activity.

To identify and further examine related patterns of specific convective activity, monthly TD time series at selected locations were correlated with those of all grid points in the investigation area. We used the Spearman rank correlation coefficient $r_s$ since it is independent of the underlying pdf and more robust against outliers compared to the widely used Pearson product-moment coefficient (Wilks, 1995). In this way, correlation maps with respect to selected reference grid cells were obtained. For the sake of better comparability, the underlying time series were smoothed by a moving-window before. We tested the results for statistical significance by applying the univariate bootstrap method, which should be preferred to bivariate bootstrapping especially for small sample sizes (Lee and Rodgers, 1998). If $n$ represents the length of the sample time series, the univariate algorithm implies randomly drawing $n$ times with replacement from each of the two samples separately. This procedure is repeated 1,000 times, leading to the null distribution of the correlation coefficient. In the sense of a two-sided test, the null hypothesis $H_0$ is rejected if the observed $r_s$ is greater than the 97.5%-quantile or less than the 2.5%-quantile of the null distribution ($\alpha = 0.05$).

### 2.2.4 NAO and convective activity

We investigated the influence of the NAO on the mean spatial distribution of convective activity in Europe by assessing the deviations of TD frequency from climatology during strongly positive (NAO > 1) and negative (NAO < -1) phases, respectively. For this purpose, the variable $D$ measuring the influence of the NAO index on convective activity was defined as

$$D_{\pm} = \frac{rf\{TD = 1 \mid NAO \gtrless \pm 1\} - rf\{TD = 1\}}{rf\{TD = 1\}}, \qquad (3)$$

with $rf\{a|b\}$ denoting the mean relative frequency of event $a$ given event $b$. Thus, mean relative frequencies obtained by averaging over the monthly TD frequency values were compared between the respective NAO phases and the climatology. The variable $D$ was calculated for each grid cell separately. A value of $D = 1$ means that thunderstorm days are twice as frequent during the respective NAO phases (positive or negative) as compared to the total sample. We assessed significance by means of a two-sided bootstrap test ($\alpha \in \{0.05, 0.10\}$) regarding the test statistic $D_{\pm}$. Here, the null hypothesis $H_0$ states that an observed relation between NAO and lightning activity within a grid box is simply due to chance.



## 3   Results

In the following, spatial and temporal patterns of TD numbers across the investigation area are discussed. More detailed analyses are presented for four example sub-regions representative of specific characteristics of convective activity: the Ticino (m in Fig. 1), Côte d'Azur (y), Maritime Alps (l), and Bavarian Prealps (t).

### 3.1   Spatial distribution of convective activity

The mean annual number of TDs presented in Fig. 3 shows a very large spatial variability. On the large scale, the distribution is dominated by a distinct northwest-to-southeast gradient, which is mainly caused by the distance to the Atlantic substantially affecting the general climate. While less than two TDs occur on average over Brittany (indicated by b in Fig. 1) and Cornwall (a), more than 20 days are found in the vicinity of the Alps. Superimposed on this overall trend are several sub-structures on the regional scale such as the local maximum downstream of the Black Forest (s) or the minimum in the upper Rhône valley (i). These sub-structures are mainly related to flow deviations, thermally-driven wind systems, and local moisture anomalies in the presence of orography.

The three primary maxima all extend along the southern Alpine range. One of them stretches across the region between the Swiss Canton of Ticino (m) and the Italian city of Turin in the southwest (up to 21 TDs). Similar values are reached in southern Austria, namely along a bow-shaped area between the eastern edge of the Upper Tauern (q) and the foothills northeast of Graz (r). In addition, high values (up to 13 TDs) are observed widespread across the Austrian Eastern Alps. The third principal maximum (up to 16 days) is located north of Nice over the Maritime Alps (l), extending, albeit weakened, northward to Lake Geneva and along the Swiss Prealps (h).

Ticino has already been identified in previous studies as one of Europe's core areas regarding thunderstorm activity (Van Delden, 2001; Schulz and Diendorfer, 2002; Nisi et al., 2016). The distinct maximum can be linked to increased potential instability due to low-level moisture originating from the Mediterranean and the water bodies in the upper Po valley (o). Instability is even further increased when cold air masses advected from the northwest and blocked by the Alps at lower levels reach Ticino aloft only (Costa et al., 2001). Convection triggering mechanisms are provided by orographically induced flow deviations, outflows from mature convective cells and catabatic-anabatic wind systems leading to low-level convergence zones (Gladich et al., 2011; Nisi et al., 2016). Over southern Austria, the high number of TDs already found by other authors (e.g., Wakonigg, 1978; Schulz et al., 2005; Punge and Kunz, 2016) can be plausibly explained by advection of potentially unstable air over the flat terrain in the southeast, which accumulates in the Graz and Klagenfurt basins and is lifted at the nearby foothills. The close proximity of the Maritime Alps (l) to the Mediterranean allows for very moist and warm maritime air getting lifted over the complex topography. Sea breezes may play a role in convection initiation as well. The lower Rhône valley (f) provides a path for potentially unstable air advected from the Mediterranean northwards. Lifting at the adjacent slopes leads to elevated numbers of TDs in that area. Advection of elevated mixed layers (EML) from the southwest often creates convection-favoring conditions along the Swiss Prealps (Peyraud, 2013).





Besides the TD maxima connected to the Alps, elevated values are also found in the vicinity of low mountain ranges or hilly terrain. In Germany, the most prominent maxima are located over the Bavarian Prealps (t; ∼15 TDs) and between Black Forest and Swabian Jura (s; ∼12 TDs). Secondary maxima can be observed over the Ore mountains (x), the Bavarian-Bohemian

Forest (u) and, unlike previously stated (Finke and Hauf, 1996), only over some parts of the hill terrain to the west, for example the Sauerland (w; 9–11 TDs). A further remarkable feature is the meridional streak of increased values along the German-Polish border (∼9 TDs). In France, the strongest extra-Alpine lightning activity is observed over the Pyrenees with highest values registered on Spanish territory in the south (c, ∼14 TDs). Other maxima are located over Massif Central (e) and the Jura Mountains (g). Remarkably, a broad streak of higher frequencies also ranges from the Bay of Biscay to the northeast.

Regarding ambient conditions, the increased TD values in Southern Germany can be attributed to a pronounced north-to-south gradient in thermal stability or convective energy (Brooks et al., 2003; Mohr and Kunz, 2013). In Southern Bavaria, convective cells are often initiated at the first Alpine range before being advected northeastward by the steering flow (Finke and Hauf, 1996; Hagen et al., 1999). The maximum between Black Forest and Swabian Jura has been attributed in previous studies to flux convergences and gravity waves forming downstream of Black Forest in conjunction with moist air advection from the

upper Rhine valley (Kunz and Puskeiler, 2010; Puskeiler et al., 2016). Similar flow effects might be relevant for other areas with elevated TD frequencies. Relatively high values along the German-Polish border peaking in the Ore Mountains can be interpreted as an extension of a High Tatras maximum found by Bielec-Bąkowska (2003), Kolendowicz (2012) and Czernecki et al. (2016). Regarding the hill country and low mountain ranges in Western Germany, orographic structures seemingly are not sufficiently marked in some cases in order to provide a significant trigger mechanism. Our results obtained for France largely

conform with the findings of others, for example Lassègues et al. (2003). EMLs originating from the Spanish plateau, denoted to as Spanish Plume by Morris (1986), frequently lead to the formation of thunderstorm tracks observed between the Bay of Biscay and Massif Central (Van Delden, 1998). Thunderstorm development over the Pyrenees (cf. Soriano et al., 2005; Santos et al., 2013) is favored by moist Mediterranean air masses being advected along the Ebro valley in Spain.

Conversely, pronounced minima with only one ore two TDs are found over Brittany (b) and Cornwall (a). Other areas with

a low TD number include the upper Rhône valley (i) northwest of Ticino, large parts of the Canton of Grisons/Switzerland (n), and the Oetz valley in Tyrol/Austria (p). In addition, two narrowly confined minima can be detected along the French-Italian border. In the upper Aosta Valley (j), only 7 TDs are observed as opposed to the 13 TDs just below the marked bend. Even lower values are present in the Guisane valley north of Briançon (k) in France.

Convective activity does not reach its minimum over glaciated mountain ranges as proposed by earlier studies (e.g. Manoochehrnia

et al., 2007), but along the neighboring deep valleys. Due to shadowing effects, the evolution of moisture flux convergences is strongly impeded there. In contrast, the presence of snow and ice cover in summit regions reduces thunderstorm frequency to a much lesser extent, as can be seen for example by the comparatively high number of TDs in the partially-glaciated Upper Tauern in Austria (q). Low TD numbers over the Brittany, and, likewise, along the Baltic Sea coast, can be explained by the stabilizing effect of the cool sea water during summer months. Note that the extension of the Breton landmass seemingly is not

sufficiently large in order to allow for a local increase of convective activity due to solar heating. In spite of higher temperatures, the Mediterranean also stabilizes the maritime boundary layer leading to a reduction in thunderstorm activity along the Côte





d'Azur (Anderson and Klugmann, 2014). It must be stressed that the Mediterranean simultaneously serves as an important source of moisture for convective cell formation over the neighboring Maritime Alps as aforementioned.

Considering lightning density instead of TDs, spatial variability is characterized by a largely similar large-scale pattern of

minima and maxima (not shown). Pronounced differences on the local scale, however, occur where TDs go along with lower or higher flash numbers compared to the mean. This effect can be studied comparing gamma distributions fitted to diurnal flash numbers among six exemplary subregions (Table 1). Recall that increasing shape and scale parameters, $\alpha$ and $k$, result in less skewness and stretching towards higher values, respectively (see Eq. 1). As expected, high flash numbers are emphasized in the distributions belonging to several hot spots of TD occurrence, such as Ticino ($k = 33.0$), Maritime Alps ($k = 25.6$) and

southern Bavaria ($k = 23.5$). Conversely, the Pyrenees, albeit representing a prominent maximum with respect to TDs as well, are characterized by a substantially lower scale parameter ($k = 10.5$). Regarding the locations of TD minima, the fitting result shows that the Brittany expectedly exhibits very low lightning incidence ($k = 5.2$), as opposed to a fairly high scale parameter at Côte d'Azur ($k = 17.0$). The anomalies of $k$ observed both for the Pyrenees and Côte d'Azur are only partially compensated for by the shape parameter. Thus, the extrema visible in Fig. 3 can be grouped according to the average lightning incidence per

TD.

## 3.2 Temporal variability on diurnal and seasonal scales

Besides the spatial variability discussed in the previous paragraph, convective activity exhibits a strong temporal variability on diurnal, seasonal, and interannual time scales. The following section investigates the two former modes of variation.

### 3.2.1 Joint annual-diurnal cycles of lightning frequency

In addition to the large spatial variability in TDs as discussed in the previous section, also the diurnal cycle of lightning frequency differs considerably across the investigation area. In the example regions, the global maxima occurring in the afternoon or evening over the Maritime Alps, Ticino, and Bavarian Prealps are distinctly shifted against each other by several hours (Fig. 4). While the maximum in the former region is registered around 14 LT, it occurs 4 hours later at 18 LT in Ticino and at 20 LT in the latter region. Furthermore, while in Ticino lightning has an elevated frequency during nighttime, it is substantially

reduced over the Maritime Alps, especially between 22 and 10 LT. Contrarily, high levels of lightning persist also throughout the night at the Côte d'Azur, yielding two broad peaks between 00 and 06 LT and between 13 and 20 LT.

Plotting mean annual cycles of TD frequency for the four example subregions (Fig. 5) likewise reveals distinct spatial features. Predominantly, convective activity exhibits a broad summer maximum accompanied by a strong increase and decrease in spring and autumn, respectively, whereas it is characterized by a sharp autumn maximum at Côte d'Azur. However, even those

regimes with frequent summer lightning differ significantly from one another, for example TD frequency increases much later in Ticino than in Southern Bavaria or the Maritime Alps, where autumn decrease in turn is delayed.

To consider both, daily and annual cycle, jointly, we used a kind of Hovmöller diagram for the four example regions showing the normalized lightning frequency as a function of the two temporal modes (Fig. 6). To better highlight the most important temporal patterns, two-dimensional smoothing of the data was performed by employing 10-day and 100-minute running means,





respectively. As can be seen, most diurnal cycles vary substantially throughout the year with large differences found also among neighboring regions such as Maritime Alps and Ticino. The former distribution (Fig. 6a) exhibits a symmetric diurnal cycle with almost vanishing nighttime activity and a maximum around 15 LT remaining approximately constant from the middle of June until the beginning of August. In Ticino, contrarily, nighttime and morning thunderstorms are quite common (Fig. 6b), but not before the end of June. The afternoon maximum is shifted into the early evening (18 LT) with the season of highest values distinctly shortened compared to the Maritime Alps. Along the Bavarian Prealps, the diurnal cycle has some similarities with that of the Maritime Alps, but it is clearly displaced towards later hours and consequently extends until after midnight (Fig. 6c). Both in Ticino and the Bavarian Prealps, the late summer drop in evening lightning frequencies (see Fig. 5) does not imply cessation also of nighttime activity. By contrast, lightning frequencies behave completely different at Côte d'Azur, where a weak afternoon maximum is visible during summer months in addition to sporadic nighttime activity, before the diurnal cycle changes entirely in September with a pronounced maximum extending from the afternoon all over the night (Fig. 6d).

Several other studies (e.g. Schulz et al., 2005; Novák and Kyznarová, 2011; Wapler, 2013) that focused on parts of our investigation area have already identified the afternoon peaks in lightning frequency as a prominent feature. Our results, however, suggest that lightning regimes behave more diversely concerning the time of the maxima and, in particular, the seasonal variation of diurnal cycles. The large discrepancies between the Maritime Alps and Ticino lightning regimes, despite both being particularly steered by orographic lifting of Mediterranean air masses, can be plausibly explained by a differing level of convective organization. In the former domain, thunderstorms preferably are initiated as single cells at the mountain slopes, which dissipate quickly in the evening hours, when radiative cooling sets in. In contrast, nighttime thunderstorms in the latter region can be attributed to convergence zones due to complex orographic features and outflows from mature cells (Gladich et al., 2011; Nisi et al., 2016) frequently leading to long-lived MCS (Morel and Senesi, 2002). Organized convection has been shown to be also important in the Bavarian Prealps by Van Delden (2001). This type of convection causes the time shift of maximum lightning frequency relative to the Maritime Alps. The seasonally-dependent influence of the Mediterranean on stability accounts for further noticeable features described above, such as the jump of Ticino lightning frequency at all times of the day found for the end of June and the distinct change in the Côte d'Azur diurnal cycle taking place in early autumn (see Sec. 3.2.2). Frequent nighttime lightning in September over the Western Mediterranean has also been found by Santos et al. (2013). In the coastal zone, land breezes might additionally favor thunderstorm activity during night.

### 3.2.2 Monthly patterns of thunderstorm days

The findings obtained in the previous subsection suggest analyzing the regional peculiarities of seasonality more in detail. For this purpose, we calculated the mean spatial distribution of TD numbers for each month separately (Fig. 7). Generally, these maps confirm the dominance of a single summertime maximum in most areas caused by the seasonality of insolation and low-level temperature. However, several pronounced local differences are visible as well. In April, convective activity is already slightly increased in those regions, where the overall primary maxima occur. Over the High Alps, TDs do not occur at all. In May and June, maximum values are observed in southern Austria (3.5 and 5 TDs, respectively) with the area of peak activity shifted southwestward. At this time, TD frequency in Ticino yet is markedly lower than in Austria, before the



Ticino maximum becomes dominant from July onwards (up to 7 TDs). Note the clear delay of springtime increase in the latter area compared to the Bavarian Prealps (cf. Fig. 5). In August, lightning activity decreases nearly everywhere except for the Côte d'Azur area. There and in the adjacent lower Rhône valley, convection even becomes more frequent in September (up to 1.8 TDs). Although TDs have turned quite rare in most regions by then, the Ticino and Pyrenees maxima are still present (2 and 2.5 TDs, respectively).

In Graz Basin, low level moisture is able to accumulate much earlier than in the Alpine areas located farther to the west. There, later snow melt additionally prevents the slopes from being heated by the sun in spring (Wakonigg, 1978; Bertram, 2000), which particularly affects the summit regions. Both factors lead to the southwestward shift of lightning activity in Austria culminating in very high TD numbers at the western edge of the Upper Tauern in July, which might be caused by moisture flux convergence at the junction of the two major Drau and Puster valleys. Whereas stabilization by the cool Mediterranean damps the Ticino maximum until June, the favorable local orographic features (cf. Sec. 3.1) lead to the extremely intense activity observed in July. This southwestward relocation of the peak convective region agrees with the results of an earlier study concerning MCS frequency (Morel and Senesi, 2002). In September, lightning events have become rather scarce because of a decrease of mean potential instability in connection with solar insolation weakening. Simultaneously, the Mediterranean meanwhile has warmed considerably relative to the air, favoring thunderstorm formation in the adjacent areas and leading to the striking rearrangement of the spatial pattern of convective activity observed. Anderson and Klugmann (2014) showed that this effect even strengthens in October, where we have got no data.

### 3.2.3 Duration of annual lightning period

The large differences in seasonality suggest investigating the duration of annual lightning period separately for each grid cell. For this, we determined the average values of those days-of-year when the first and last TD occur, respectively (Fig. 8).

Both patterns of first and last TD occurrence are connected to the mean spatial distribution of TDs (Fig. 3) in the sense that a high/low local TD number usually implies a longer/shorter lightning period. Examples for this relation include the pronounced northwest-to-southeast gradient, and the discrepancy between the Ticino-Turin region (end of April until beginning of September) and the nearby upper Rhône valley (end of June until first half of August). Apparently, locations favoring the development of convective cells are in many cases characterized by a long persistency of these conditions as well. However, lightning regimes behave in a more complex way in several regions. For instance, orographic features are attenuated considering the distribution of the last TD recording (e.g. Maritime Alps, upper Rhône valley and northern Alpine range), which, in addition, bears much less resemblance with the mean pattern of convective activity than the distribution of the first TD. Instead, that pattern is characterized by a broad area in Southern France, where TDs occur widely spread late in the year, particularly in the surroundings of the lower Rhône valley near to the Mediterranean coast (middle of September). Furthermore, the convective period over the Mediterranean itself starts late, but lasts quite long with the last TD observed in the first half of September.

Three steering factors already mentioned in Sec. 3.2.2 trigger these pronounced differences: persistent snow cover, lack of low-level moisture and the annual cycle of sea surface temperature (SST) of the Mediterranean. Since the two former conditions strongly impede thunderstorm formation in the High Alps and the adjacent deep valleys until late spring (cf. Fig. 7 a,b),





convective season is delayed in those regions compared to their forelands, leading to the strong gradients observed in the first TD distribution (Fig. 8, left). The role of SST, already identified as one of the main drivers of annual TD cycle (see Fig. 5 and 7), is reflected by the very late cessation dates occurring north of the French coast. Note in particular that the reduced thermal stability connected to high SST allows for late last TDs everywhere in that region, independent of orographic structures prevailing.

### 3.3   Interannual variability

Even though a period of 14 years, where lightning observations are available, is far below a climatological time span (e.g., 30 years), it is sufficient to study interannual variability in addition to the diurnal and seasonal cycles discussed above. However, owing to the relatively short time series, the main focus is put on the spatial scales of convective activity and the potential relation to teleconnections.

#### 3.3.1   Variability of annual TD numbers

Annual lightning totals vary substantially during the time period considered. In the year of 2006, for example, a total of 3.6 million lightning flashes were recorded, whereas convective activity was rather low with approximately 2.1 million flashes in 2010 and 2012. Simultaneously, complex spatial differences regarding interannual variability are present. Although there are some years with increased or reduced TDs nearly everywhere, such as in 2006 or 2010, respectively (Fig. 9 a,b), frequencies usually do not behave spatially consistently. One example is the pattern observed for the year of 2001, when the Ticino-Turin maximum already discussed above has developed well in contrast to low values in southern Austria and over the Pyrenees. Other distinct local maxima such as those of the Paris basin (d in Fig. 1) or around Hamburg (v) are not present in the 14-year mean (cf. Fig. 3). In 2013, contrariwise, the Pyrenees and Maritime Alps exhibit increased convective activity as opposed to Ticino and southern Austria.

The large annual and interannual variability of lightning frequency can be attributed – at least partly – to prevailing upper troposphere flow patterns favoring or preventing thunderstorm development. In addition, as shown by Piper et al. (2016), days with large-scale weather types favorable for convection frequently form clusters of several days, i.e. the incidence of such weather types implies a considerable probability for a convective situation lasting for a longer time period. This persistence effect amplifies the number of the corresponding favorable flow patterns in those years, when they frequently set in, and hence, the interannual variability of convective activity.

Another aspect of interannual convective variability is the dispersion of TD numbers, which can be assessed for each grid cell using the coefficient of variation (CV; Sec. 2.2.3). Notably, large parts of the investigation area exhibit a fairly homogeneous CV pattern with values around 0.4 (Fig. 10), taking into account a considerable background noise of $\pm 0.1$. Thus, no significant spatial fluctuations of dispersion is visible here, in spite of the spatial peculiarities regarding year-to-year variability discussed above. Nonetheless, CV values are higher in areas where mean TD numbers are particularly low, and vice versa. For example, higher values at several grid points are obtained for regions with infrequent lightning such as Northern Germany. Note, however, that higher values and larger gradients of CV may be caused by small integer numbers, where an increase of TD frequency by





only 1 day in a specific year yields a stronger increase when the number of TDs is low. This effect culminates over the Atlantic with values up to 3.3, where CV approaches its singularity.

### 3.3.2 Spatial correlations

According of the findings described in the previous subsection, interannual variability is only partially coupled among different regions. Comparing correlation maps for four different reference points (Fig. 11) shows that the peripheries, inside of which the time series tend to cohere, largely vary in terms of area and shape. The grid cell located between the Black Forest and Swabian Jura (Fig. 11a) exhibits significant correlations with a vast region extending from the French Alsace to the easternmost parts of Austria. Conversely, the area of correlated grid cells strongly decreases when the reference cell is set to a location northeast of

the Ore Mountains (Fig. 11b). Here, significant correlations lie inside of a narrow band ranging from the western edge of the Ore Mountains along the Czech and Polish border all the way northward to the Baltic Sea coast. When looking at correlations with respect to the southern Pyrenees (Fig. 11c), we find high values not only along the main axis of the Pyrenees, but also inside of a detached area basically comprising the Maritime Alps. By contrast, the correlation pattern obtained with respect to a location within the Ticino-Turin lightning maximum (Fig. 11d) features significant values only inside a closely confined region.

In particular, there is no correlation even with large parts of the Ticino itself. Negative values are visible within a remote region, comprising for example grid points in northwestern Spain. The relatively large area in the Netherlands exhibiting significant positive values might be explained either by a complex spurious correlation or, more realistically, by random effects despite the high significance level employed.

  From the examples shown in Fig. 11, it can be concluded that the large-scale flow configurations leading to convection-

favoring conditions and sufficient lifting substantially differ among the various regions. For example, the relation of the temporal variability in TD numbers between the Pyrenees and the Maritime Alps points to a large relevance of weather types causing Mediterranean air to impinge on the respective mountain slopes. Strikingly, the latter region is not correlated with the nearby Ticino-Turin lightning maximum at all, where strong meso-scale correlation coefficient gradients show that even neighboring locations may behave inconsistently given a specific large-scale flow situation prevailing, presumably due to the complex

orography. Large correlations along the German-Polish border similar to the area of the local lightning maximum (Fig. 3) point to the influence of potentially instable air originating from the southeast in connection with the southern Polish lightning maximum addressed in Sec. 3.1. Flow patterns advecting warm and moist subtropical air masses to Southern Germany, where they spread to the east and persist in a large region over a longer period of several days, might cause the interrelation of the time series in this area. However, low-level moisture and instability are also generated locally by means of evaporation and

solar heating, yielding another contribution to the interannual variability observed. Note furthermore that convection may be triggered, in addition to large-scale lifting as discussed above, by processes taking place in the boundary layer and depending on local characteristics such as soil moisture and land use. Differentiating the impact of large-scale processes on interannual variability from those local factors might yield further insights. However, this aspect is beyond the scope of this paper.



### 3.3.3 Relationship between convective activity and the NAO

The complex characteristics of interannual variability found in the time series of TDs suggest investigating whether a systematic relation can be established between convective activity and atmospheric teleconnection patterns in terms of the NAO index, which is of relevance for European weather and climate (e.g. Della-Marta et al., 2007; Hurrell and Deser, 2010).

As can be clearly seen in Fig. 12, there is a distinct relation between NAO phases and lightning activity. Over most of the area, TD frequency is considerably increased during strongly negative NAO phases ($N_-$; Fig. 12 c,d). A prominent feature, for example, is the absolute maximum in eastern Austria, where TD frequency doubles in places ($D_- = 1$) compared to the entire sample. Statistically significant positive values of $D_-$ are reached in several other areas, for example in the southern half of Germany and along the German-Polish border. However, a sharply delineated zone of near-zero, negative $D_-$ values is located inside the margins of the French Alps comprising all the mountain ranges between the Maritime Alps and Lake Geneva. Within a marine area, which can be perceived as a southward extension of this region, even significant negative values are observed.

Positive NAO phases ($N_+$) predominantly go along with a decrease of TD frequency (Fig. 12 a,b). The most striking feature is a zone of strong reduction stretching from the Mediterranean coast in northwestern Italy along the French-Italian border northward to the upper Rhône valley in Switzerland with an eastward extension to Grisons and Tyrol (cf. Fig. 1). Note that the Ticino-Turin lightning maximum situated in between is characterized by a weaker decrease. Other regions, where $D_+$ exhibits significantly negative values are the lower Danube valley in northeastern Austria, the northern foreland of the Pyrenees, and parts of the North German Plain. The almost contiguous zone of highly significant results extending from the western Bay of Biscay to Cornwall suggests the area-wide suppression of convection ($D_+ = -1$) being a reliable observation instead of being a statistical artifact due to the small event sample sizes in that region. Comparing the locations of the $D_+$ minima to the mean spatial pattern of TDs (Fig. 3) leads to the observation that most of them are characterized by weak average convective activity.

Since the NAO index considered in this paper has been defined by the leading mode of a PCA performed separately for each month (see Sec. 2.1.2), the centers of action change their position throughout the year. Thus, we cannot interpret the positive NAO phase as a period of strong zonal flow in Central Europe, as would be the case using the classical normalized pressure difference between two points. Instead, the southern center of action is given by a positive/negative anomaly band stretching from the United States to Europe when the NAO index is positive/negative, with its latitude oscillating between 30 and 35°N in winter between 40 and 50°N in summer (Barnston and Livezey, 1987; Hurrell and Deser, 2010). Consequently, during SHY, negative phases $N_-$ go along with high geopotential gradients over southern Central Europe, whereas positive phases $N_+$ correspond to a jet stream shifted to Northern Europe. For $N_-$, short-wave troughs frequently affect the investigation area and provide, primarily in the southern and eastern parts, quasi-geostrophic forcing, which is conducive to convective initiation. This mechanism might explain the increase in TD frequency connected to $N_-$. The contrary decrease of TD frequency over the French Alps confirms the exceptional nature of this lightning regime (cf. Sec. 3.2.1). In the case of $N_+$, the lack of lifting as a consequence of the positive geopotential anomaly located over Central Europe yields weaker convective activity. Furthermore, strong capping inversions inside of associated high pressure systems might additionally inhibit cell formation. Both factors seem to be detrimental especially in those areas where convective activity is low on average, such as over the marine areas



around Brittany. Another example is the upper Rhône valley, where ambient conditions have shaped up to be rarely conducive to convection due to shadowing effects (see Sec. 3.1), and a favorable flow pattern therefore is indispensable for thunderstorm development. In contrast, complex lifting mechanisms in nearby Ticino allow for convection in spite of the absence of large-
scale forcing.

## 4    Conclusions

The spatio-temporal variability of convective activity has been investigated within a study domain comprising large parts of western Europe based on 14 years of lightning data (April – September). For this purpose, we developed an objective definition of the dichotomous variable thunderstorm day (TD), which is robust in the case of single severe events and simultaneously
neglects small-scale weak thunderstorms. We studied the mean spatial distribution of annual TD numbers, compared diurnal and seasonal cycles of lightning incidence among several European sub-regions, and performed analyzes of interannual variability. In particular, the impact of the NAO on convective activity was investigated, since it represents a manifestation of the large-scale flow configuration.

It was found that the mean spatial pattern of thunderstorm activity is characterized by a pronounced northwest-to-southeast
gradient between very low values ($\sim$2 TD pear year) observed in northwestern France and strong maxima ($\sim$21 TD) in some parts of the southern Alps, for instance in the Swiss Canton of Ticino. Superimposed on this large-scale trend are several distinct regional structures such as the pronounced lightning minima over the deepest Alpine valleys. Seasonality is characterized by a single maximum in July in most places. However, the area of maximum convective activity moves in a southwestward direction from southeastern Austria in April to the surroundings of Ticino in July. At the French Mediterranean coast, contrarily,
thunderstorm activity does not reach its maximum until September. Regional diurnal cycles of flash frequency mostly exhibit a single afternoon or evening maximum, with its exact time varying substantially. Distinct spatial differences are present regarding the occurrence of nighttime thunderstorms. Moreover, most diurnal cycles feature pronounced seasonal changes, such as along Côte d'Azur in September, when a transition takes places from a regime dominated by a weak afternoon maximum to frequent nighttime lightning.

Multi-annual TD time series are spatially interrelated to a limited extent only. Correlation maps show that the area exhibiting a high and significant degree of correlation varies strongly in size for different reference points, and can be extremely small in some cases. The NAO has a significant impact on lightning probability with its negative/positive phase generally favoring/reducing convective activity. In some areas, the strength of this effect depends on the orographic structures prevailing.

Three main factors governing the spatio-temporal variability as described above are given by the variable distance to marine
areas, local orographic features leading to flow deviations and, consequently, convergence zones, and regional differences in the abundance of low-level moisture. During summer half-year, the Atlantic, North Sea, and Baltic Sea represent sinks of low-level sensible heat stabilizing the atmospheric boundary layer. The Mediterranean, however, additionally provides a source of latent heat due to the higher temperatures prevailing. Therefore, thunderstorms are inhibited over the Mediterranean coastal zone during summer, while they are promoted farther inland, where the humid air impinges on the Alps. In September, the



Mediterranean has become warm relative to the air leading to more unstable conditions over the water, also during nighttime. Complex local flow patterns in combination with moist low-level air allow for distinct convective maxima along the Alps and some low mountain ranges, for example downstream of Black Forest. The local orographic structures also affect the degree of convective organization relevant in the context of diurnal cycles, as long-lived convective systems often are responsible for substantial nighttime activity. However, thunderstorm activity diminishes strongly, where orographic shadowing effects imply a reduction of moisture. Differing humidity levels along the southern Alpine range during early summer cause the shift of the peak convective area observed in this region.

The analysis of interannual variability shows that the steering factors of convective activity described above are not sufficient in order to explain some of the aspects observed. Instead, time series of annual lightning incidence seem to depend on specific large-scale drivers prevailing frequently in some years and inducing the advection of potentially unstable air and lifting at higher troposphere levels in those regions, where high TD numbers are observed. Our results suggest that different large-scale conditions might favor convection in the various sub-regions. The substantial impact of the NAO can be explained by its negative phase being connected to flow patterns that are associated with frequent short-wave troughs providing lifting over Central Europe, whereas during positive phases a high geopotential anomaly tends to suppress convection.

A potential weakness of our research is that the 14-year time series of lightning data does not reach the climatological time range. However, this sample should be sufficient for reliable analyzes regarding spatio-temporal variability on diurnal and seasonal time scales. As opposed to earlier studies, which all are based on shorter time spans, we were able to additionally investigate multi-annual modes of variability using well-suited significance tests in order to ensure that no misleading conclusions are drawn due to the limited sample size. Since the LF detection range the EUCLID network is based on provides reliable measurements in the case of CG flashes only, no conclusions can be inferred from our results regarding CC flashes. However, although a CC climatology might potentially reflect further meteorological aspects, the threat electric discharges pose to the society is primarily associated with flashes reaching the ground. This justifies neglecting CC lightning, particularly in view of the high instrumental demands of VHF systems (e.g. Drüe et al., 2007; Pohjola and Mäkelä, 2013).

The clear and significant impact of the NAO pattern on convective activity in Europe motivates us to scrutinize the crucial role of large-scale flow in future research. The objective is to gain further insight into the physics behind the complex spatio-temporal variability discussed in this paper. This involves evaluating additional teleconnection modes obtained by Barnston and Livezey (1987) such as the East Atlantic Pattern. To generate longer, multi-decadal time series allowing for further investigations such as trend and spectral analysis, we have additionally implemented objective weather types as indicators for a high convective predisposition (Piper et al., 2016).

*Acknowledgements.* The study was undertaken and financed under the framework of the Regional Climate Initiative REKLIM of the Helmholtz Association. We gratefully thank Siemens AG for providing lightning data and NOAA for providing NAO data.



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





**Table 1.** Gamma distribution shape ($\alpha$) and scale ($k$) parameter values with respect to daily flash numbers in various regions. For better comparability, flash numbers have been spatially averaged over the single grid cell values.

| region | $\alpha$ | $k$ |
|---|---|---|
| Ticino | 0.086 | 33.0 |
| Maritime Alps | 0.092 | 25.6 |
| Bavarian Prealps | 0.084 | 23.5 |
| Côte d'Azur | 0.050 | 17.0 |
| Pyrenees | 0.113 | 10.5 |
| Brittany | 0.035 | 5.2 |



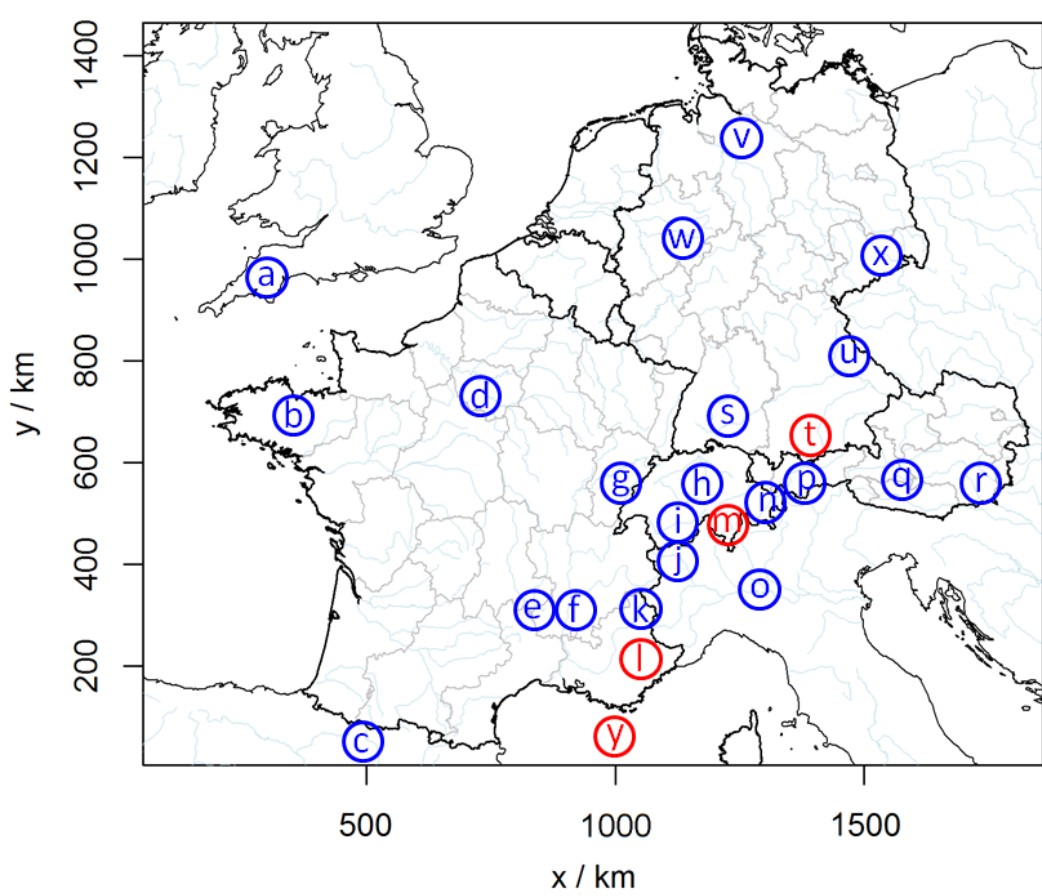

**Figure 1.** Study domain with regions marked by characters that are referred to in this study: (a) Cornwall, (b) Brittany, (c) southern Pyrenees, (d) Paris Basin, (e) Massif Central, (f) lower Rhône valley, (g) Jura Mountains, (h) Swiss Prealps, (i) upper Rhône valley, (j) Aosta valley, (k) Guisane valley north of Briançon, (l) Maritime Alps, (m) Ticino, (n) Grisons, (o) Po valley, (p) Oetz valley in Tyrol, (q) Upper Tauern, (r) Graz Basin, (s) Black Forest – Swabian Jura, (t) Bavarian Prealps, (u) Bavarian-Bohemian Forest, (v) North German Plains, (w) Sauerland, (x) Ore Mountains, and (y) Côte d'Azur. Those regions studied more in detail are highlighted in red color.





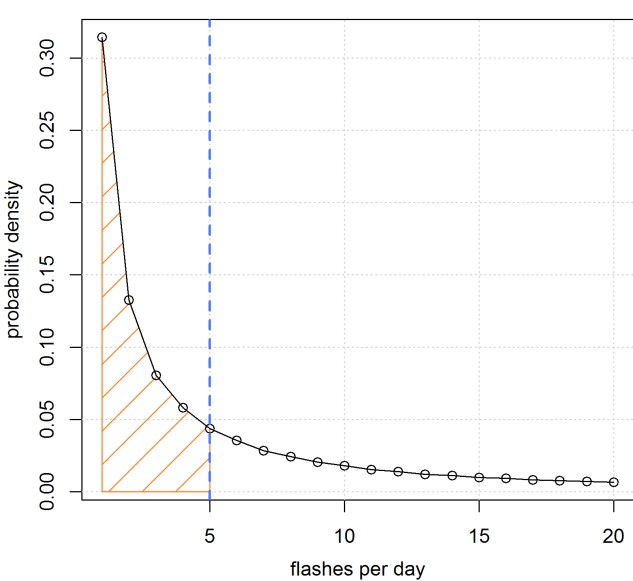

**Figure 2.** Empirical probability density distribution of daily flash numbers averaged over all grid cells within the entire investigation area during the period 2001 to 2014. Indicated are the separator between the regimes TD (yes – no; vertical dashed line) and the probability for a day with lightning that is not classified as TD (orange hatched area).





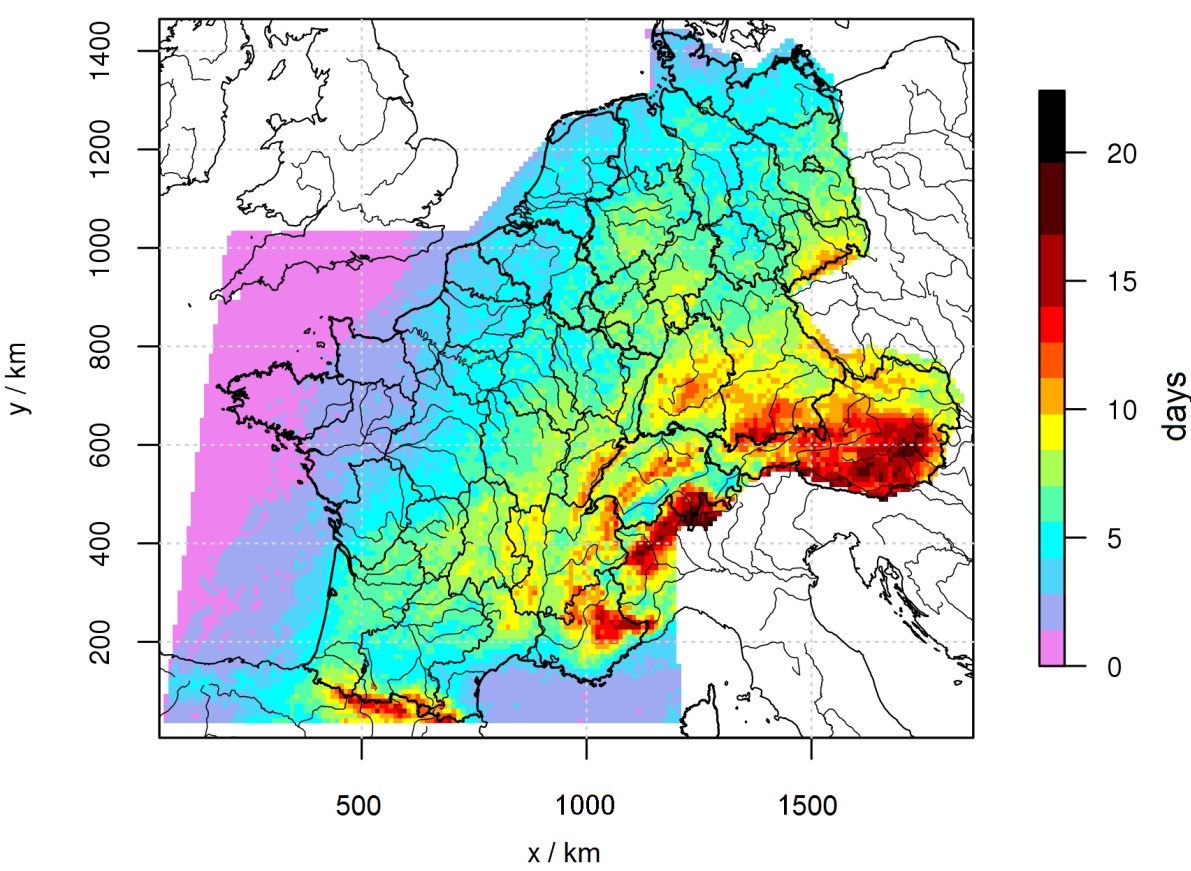

**Figure 3.** Mean annual number of thunderstorm days during the summer half-years 2001 – 2014.




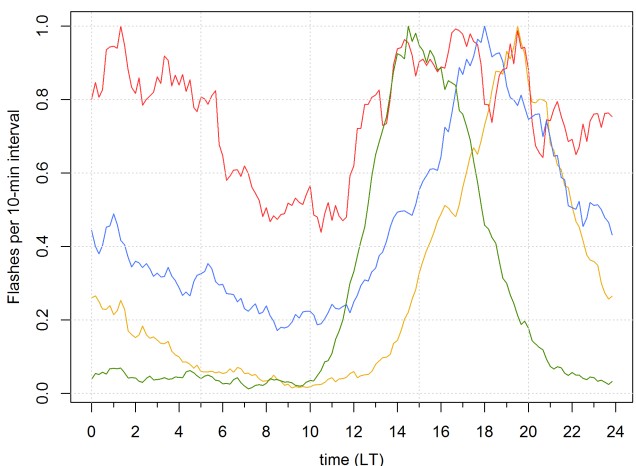

**Figure 4.** Average diurnal cycle of 10-minute flash numbers, normalized by the respective maximum, for the 4 regions: Ticino (blue), Côte d'Azur (red), Maritime Alps (green), Bavarian Prealps (yellow); time has been converted to local time (LT).

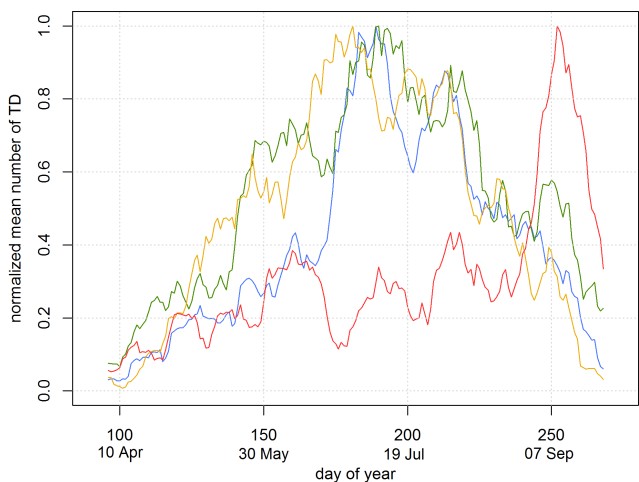

**Figure 5.** Mean relative TD frequency as 10-day moving average within intervals centered at the indicated days of year for the regions Ticino (blue), Côte d'Azur (red), Maritime Alps (green), and Bavarian Prealps (yellow). Each curve has been normalized by its respective maximum.





**Figure 6.** Average flash number as a function of local time (10-minute intervals) and day of year, normalized by the maximum values for the regions Maritime Alps (a), Ticino (b), Bavarian Prealps (c) and Côte d'Azur (d). Data have been smoothed applying a 10-day and 100-minute running mean.



**Figure 7.** Mean number of thunderstorm days during the summer half-years 2001 – 2014 in April (a), May (b), June (c), July (d), August (e), and September (f).





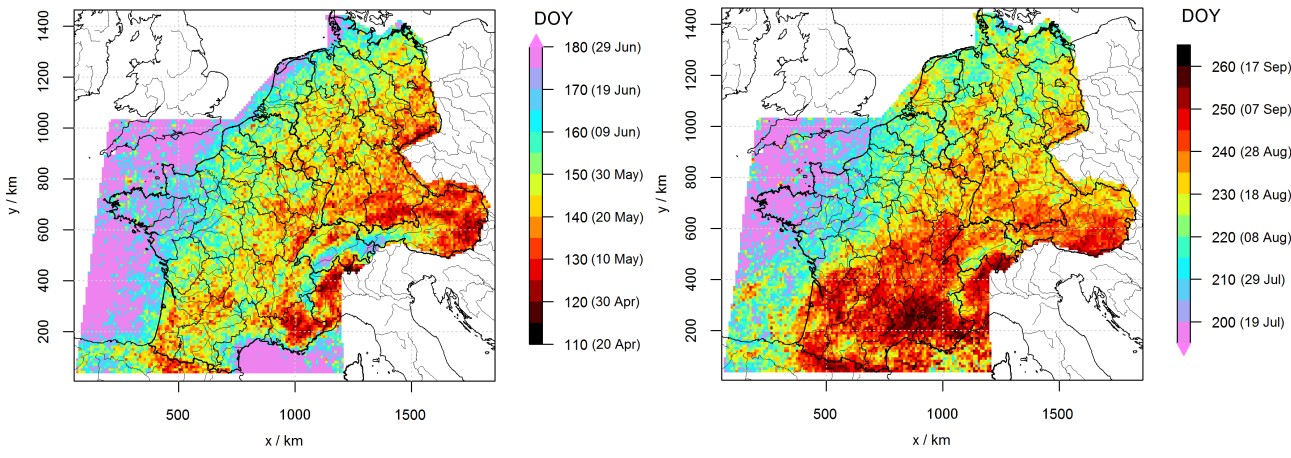

**Figure 8.** Mean day-of-year when the first (left) and last (right) thunderstorm day occur, respectively.



**Figure 9.** Annual number of thunderstorm days during the summer half-years 2006 (a), 2010 (b), 2001 (c), and 2013 (d).





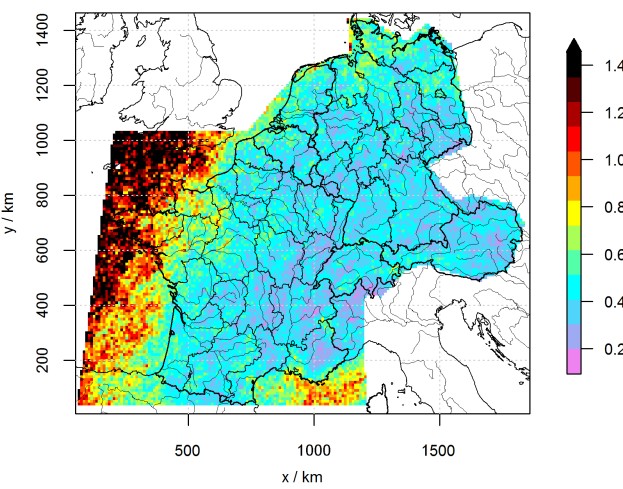

**Figure 10.** Dispersion of the annual number of thunderstorm days measured by the coefficient of variation.



**Figure 11.** Spatial correlation (Spearman's coefficient $r_s$) of the annual numbers of thunderstorm days with respect to different reference grid cells (indicated by a gray dot in each map) situated between the Black Forest and Swabian Jura (a), north of the Ore Mountains (b), on the southern side of the Pyrenees (c), and southwest of Ticino (d). Values that are not statistically significant (Si = 95%) have been set to zero.



**Figure 12.** Relative deviation (D) of the monthly number of thunderstorm days calculated with respect to months with an NAO index greater than +1 ($N_+$) from that calculated with respect to all months (a). Results of a bootstrap significance test with green/yellow grid boxes corresponding to areas where the null hypothesis has been rejected with Si = 95% and Si = 90%, respectively, and red boxes denoting acceptance of null hypothesis (b). The same applies to (c)/(d), but D is evaluated with respect to an NAO index below -1 ($N_-$).