# Peer review of "Spatio-temporal variability of lightning activity in Europe and the relation to the North Atlantic Oscillation teleconnection pattern"

_Natural Hazards and Earth System Sciences, 2017_

## Referee Comment (RC1) · Anonymous Referee #1 · 24 Feb 2017

The paper describes nicely the relation between lightning and NOA (North Atlantic Oscillation). Still, the authors may comment and adjust the paper in several aspects: 1. Why is only a 6-month summer period considered? The data is available for full years. Winter months could give interesting connections between, e.g., lightning and temperature of land versus water. 2. The seasonal shift of pressure centers (Azores, Iceland) is not clearly referenced, relating to the variation of lightning in its dependence on seasonal effects. This would help to better distinguish between local and large scale effects on lightning. It concerns also the relevance of local wind-fields in summer time that are more important than NOA. 3. The analysis of the results for lightning–

NOA correlation appears a bit short. For example, one wants more information about suppression of convection due to NOA+ effects. 4. The climatological question, to what extent 14 years lightning and 30 years NOA fit together may need some more comments. 5. For a long time flash density maps are produced that take into account all flashes in certain grid cells. It may be interesting to learn to what extent the (total) densities correlate with the TD cells. 6. Chapter 3.1 does not present much new insight and could be shortened; too many facts are detailed that are well known. Page 1; Line 16: it is mentioned that large natural hazards occur in southern Germany; the authors may recall that there have been very extensive hail disasters also in northern Germany (2013). Page 8; Line 9: the k scale parameter is mentioned after Eq.1, but now quantities are given and the reader has no good idea what the numbers mean in a meteorological sense. Page 15; Line 23: the authors suggest that cloud lightning could be detected only with VHF methods. This is incorrect. There are systems in the US (several) and in Japan (BOLT), as well as LINET in Europe and elsewhere, which can report sufficient cloud strokes in the VLF/LF range that relate to severe weather, especially hail.

The paper could be published after the indicated suggestions have been duly considered.

---

## Referee Comment (RC2) · K. Lagouvardos (Referee) · 22 Apr 2017

Spatio-temporal variability of lightning activity in Europe and the relation to the North Atlantic Oscillation teleconnection pattern

by David Piper and Michael Kunz

Submitted to NHESSD

This paper presents an interesting study based on the analysis of 14-year lighting data over a part of central/western Europe. The paper is well written and of interest for

many readers. I would like to mention that the major part of the paper consists of a description of the temporal and spatial distribution of lightning (especially Section 3.1 is a rather long juxtaposition of locations with low/high lightning activity), while the discussion about NAO (although stated explicitly in the title of the paper) is not fully exploited. I recommend however publications of the paper, taking into account the following remarks.

Specific remarks

1. My main concern is the robustness of lighting data: the authors do not provide information about the location error and to the detection efficiency of the observing network. The latter information is very important to the reader in order to have a clearer idea on how the selection of 5 lighting flashes is justified in order to characterize a TD. Moreover, if the network experienced significant changes/modification through the elapsed 14 years (e.g. adding new sensors and/or applying modification to the location algorithms) these changes can jeopardize the robustness of results. Finally no information is given on the transformation of strokes to flashes (although I do not understand the necessity of such a transformation).

2. Reference to previous work on lighting climatology: I bring to authors' attention the recent publication of Kotroni and Lagouvardos (2016) (Lightning in the Mediterranean and its relation with sea-surface Temperature, Environmental Research Letters, 2016) which comprises a 10-year lightning climatology over a major part of Europe. Therefore the authors should modify accordingly their remarks in p2, lines 16-17. In this publication you can also find a discussion on the relationship of SST with lightning, an issue that is also mentioned in your paper.

3. The analysed area lacks a part of NE Italy and Slovenia, areas being identified by previous studies have as the hot-spots of lightning in Europe (Anderson and Klugmann, 2014; Kotroni and Lagouvardos 2016), is there any reason for that?

4. In Section 3.1, p6, line 12: Which is the meaning of"local moisture anomalies"?

5. In the discussion in Section 3.3.2: how the authors believe that different vegetation types can influence the correlation between regions? Since many studies in the past have discussed this issue, I would suggest the authors having a look on this.

6. As stated in the beginning, Section 3.3.3 devoted to the relation with NAO is not satisfactorily developed. Since this aspect of investigation is original, one would expect a more thorough discussion, maybe based on the analysis of other upper-level meteorological fields. In any case, I strongly believe that additional work on this issue would result in a more solid publication. In the light of the same remark, I would suggest further refining the last sentence of the abstract and a more comprehensive concluding part (in the conclusion section, only 3 lines are devoted to NAO relation to lightning).

––––––––––––––––––––––––––

---

## Author Response (AR1)

Dear Referees,

thank you very much for your work and the useful and valuable comments on how to improve the scientific quality of our manuscript. Please find below our reply to the individual points, marked with an "AC" (author's comment).

Best regards,
David Piper and Michael Kunz

Response to the referee comments: Referee #1:

1. Why is only a 6-month summer period considered? The data is available for full years. Winter months could give interesting connections between, e.g., lightning and temperature of land versus water.

AC: We decided to confine our analyses to the summer half-year, because these months represent the main convective season in most parts of our investigation area. However, we agree that the spatio-temporal variability of winter lightning is an interesting issue as well. Due to the much smaller event samples in this case, longer time series would be necessary for statistically reliable studies.

2. The seasonal shift of pressure centers (Azores, Iceland) is not clearly referenced, relating to the variation of lightning in its dependence on seasonal effects. This would help to better distinguish between local and large scale effects on lightning. It concerns also the relevance of local wind-fields in summer time that are more important than NOA.

AC: It would be interesting to analyze the relation between convective activity and the NAO for each month separately. In order to obtain reliable results, however, longer time series of lightning data are necessary. Since the NAO anomaly patterns can be viewed as constant by approximation within summer half-year, we decided not to differentiate between the individual months.

3. The analysis of the results for lightning–NOA correlation appears a bit short. For example, one wants more information about suppression of convection due to NOA+ effects.

AC: We will add two figures showing the anomaly patterns of upper-level wind field and $\theta_e$, respectively, in order to clarify the reasons of convection suppression during NAO+.

4. The climatological question, to what extent 14 years lightning and 30 years NOA fit together may need some more comments.

AC: We used a subsample of the NAO time series that is defined by the time period lightning data were available. We will clarify this aspect in the text.

5. For a long time flash density maps are produced that take into account all flashes in certain grid cells. It may be interesting to learn to what extent the (total) densities correlate with the TD cells.

AC: Since the basic features of flash density and TD frequency maps are fairly similar to one another, we decided not to include an additional flash density figure in the manuscript. We will add an explanatory sentence about that.

6. Chapter 3.1 does not present much new insight and could be shortened; too many facts are detailed that are well known.

AC: We will check where it is possible to abridge this section and where the findings can be summarized. However, our objective is to give a comprehensive overview of thunderstorm activity in the large investigation area considered. Given the fact that previous studies focused both on much smaller domains and shorter time series we think that this section provides significant scientific added value.

Page 1; Line 16: it is mentioned that large natural hazards occur in southern Germany; the authors may recall that there have been very extensive hail disasters also in northern Germany (2013).

AC: This is correct. However, different hail climatologies have shown that the most prominent hail hot spots are located in the southern parts of Germany (e.g. Punge et al., 2014, Puskeiler et al., 2016); we will add a comment on this.

Page 8; Line 9: the k scale parameter is mentioned after Eq.1, but now quantities are given and the reader has no good idea what the numbers mean in a meteorological sense.

AC: We will explain this more in detail.

Page 15; Line 23: the authors suggest that cloud lightning could be detected only with VHF methods. This is incorrect. There are systems in the US (several) and in Japan (BOLT), as well as LINET in Europe and elsewhere, which can report sufficient cloud strokes in the VLF/LF range that relate to severe weather, especially hail.

AC: You are right. However, the EUCLID network exhibits a much lower detection rate for IC flashes compared to CG flashes (e.g. Pohjola and Mäkelä, 2013).

The paper could be published after the indicated suggestions have been duly considered.

Response to the referee comments: Referee #2:

This paper presents an interesting study based on the analysis of 14-year lighting data over a part of central/western Europe. The paper is well written and of interest for many readers. I would like to mention that the major part of the paper consists of a description of the temporal and spatial distribution of lightning (especially Section 3.1 is a rather long juxtaposition of locations with low/high lightning activity), while the discussion about NAO (although stated explicitly in the title of the paper) is not fully exploited. I recommend however publications of the paper, taking into account the following remarks.

AC: One objective of our study is to provide a deep and comprehensive analysis of lightning activity. Due to the large investigation area compared to previous studies, we are able to perform comparisons among various regions regarding several aspects of convection such as, e.g., the seasonality of diurnal cycles. Owing to the long time series of lightning data available, we are able to also investigate some aspects of multiannual variability yielding new insights regarding the spatio-temporal behavior of convective activity. However, we will check where to abridge especially section 3.1. We also agree in broadening the discussion about the link between convection and the NAO.

My main concern is the robustness of lighting data: the authors do not provide information about the location error and to the detection efficiency of the observing network. The latter information is very important to the reader in order to have a clearer idea on how the selection of 5 lighting flashes is justified in order to characterize a TD. Moreover, if the network experienced significant changes/modification through the elapsed 14 years (e.g. adding new sensors and/or applying modification to the location algorithms) these changes can jeopardize the robustness of results. Finally no information is given on the transformation of strokes to flashes (although I do not understand the necessity of such a transformation).

AC: We will add information about detection efficiency and location accuracy. There have been no significant changes during the investigation period. The grouping procedure transforming strokes into flashes is performed internally by EUCLID. We will add a sentence about that.

Reference to previous work on lighting climatology: I bring to authors' attention the recent publication of Kotroni and Lagouvardos (2016) (Lightning in the Mediterranean and its relation with sea-surface Temperature, Environmental Research Letters, 2016) which comprises a 10-year lightning climatology over a major part of Europe. Therefore the authors should modify accordingly their remarks in p2, lines 16-17. In this publication you can also find a discussion on the relationship of SST with lightning, an issue that is also mentioned in your paper.

AC: We will add this publication to the literature cited in the discussion. However, we would like to remark that the analyses performed by Kotroni and Lagouvardos (2016) are based on VLF lightning data exhibiting a fairly low location accuracy of ~6.8 km compared to ~100 m in the case of EUCLID (Schulz et al., 2016). Using EUCLID data, we are therefore able to provide new insights regarding local-scale features, and, owing to the large investigation area, to simultaneously perform comparisons among different regions.

The analysed area lacks a part of NE Italy and Slovenia, areas being identified by previous studies have as the hot-spots of lightning in Europe (Anderson and Klugmann, 2014; Kotroni and Lagouvardos 2016), is there any reason for that?

AC: Unfortunately, we were not able to get data of lightning in Italy and Slovenia. An exception is the northwesternmost parts of Italy, which are covered by the Swiss and French datasets.

In Section 3.1, p6, line 12: Which is the meaning of "local moisture anomalies"?

AC: Negative local moisture anomalies are present, when the local orographic features inhibit low-level moisture transport into some areas, which are given by several deep and contorted alpine valleys. These moisture anomalies imply the absence of strong moisture flux convergences necessary for thunderstorm development.

In the discussion in Section 3.3.2: how the authors believe that different vegetation types can influence the correlation between regions? Since many studies in the past have discussed this issue, I would suggest the authors having a look on this.

AC: We will add some information about this aspect.

As stated in the beginning, Section 3.3.3 devoted to the relation with NAO is not satisfactorily developed. Since this aspect of investigation is original, one would expect a more thorough discussion, maybe based on the analysis of other upper-level meteorological fields. In any case, I strongly believe that additional work on this issue would result in a more solid publication. In the light of the same remark, I would suggest further refining the last sentence of the abstract and a more comprehensive concluding part (in the conclusion section, only 3 lines are devoted to NAO relation to lightning).

AC: We will additionally discuss the anomaly patterns of the upper-level wind field and $\theta_e$ for both NAO phases, which clarify the relation between the NAO and convective activity. Accordingly, we will also modify the abstract and the conclusion section.

[revised manuscript text omitted]

Oscillation (NAO) index. Those data were provided by the U.S. National Oceanic and Atmospheric Administration (NOAA). The calculation of the index values is based on rotated S-mode principle component analysis (PCA; Richman, 1986) applied to monthly mean standardized 500 hPa height anomalies (Barnston and Livezey, 1987; Hurrell and Deser, 2010) obtained from the National Centers for Environmental Prediction – National Center for Atmospheric Research reanalysis (NCEP/NCAR1; Kalnay et al., 1996). The index time series is available from 1950 onwards. However, the relation between thunderstorm activity and the NAO is studied based on a subsample of this time series, which is given by the period from 2001 to 2014 (SHY).

**2.1.3 Reanalysis data**

The output of the global reanalysis model NCEP/NCAR1 is used to examine the dynamical and thermodynamical effects of the NAO phases on convective activity. The NCEP/NCAR1 reanalysis is available for 17 pressure levels and exhibits a spatial resolution of $2.5° \times 2.5°$. For the analyzes performed in this paper, model fields of the wind vector and the equivalent potential temperature at the 300 hPa and 850 hPa pressure levels, respectively, are evaluated. Only data sets at 1200 UTC are considered since they best mirror the prevailing convective conditions. NCEP/NCAR1 data are available from 1948 onwards. However, the analyzes are limited to the time period from 2001 to 2014 (SHY), for which lightning data are available.

**2.2 Statistical methods**

**2.2.1 Binary measure of convective activity: Thunderstorm day**

[revised manuscript text omitted]

30 abundance of low-level moisture. Instability is even further increased when cold air masses advected from the northwest and

blocked by the Alps at lower levels reach Ticino aloft only (Costa et al., 2001). Convection triggering mechanisms are provided by orographically induced flow deviations, outflows from mature convective cells and catabatic-anabatic wind systems leading to low-level convergence zones (Gladich et al., 2011; Nisi et al., 2016). Over southern Austria, the high number of TDs already found by other authors (e.g., Wakonigg, 1978; Schulz et al., 2005; Punge and Kunz, 2016) can be plausibly explained by ad-

5  vection of unstable air from the southeast and subsequent lifting at the nearby foothills. Regarding the Maritime Alps (l), their close proximity to the Mediterranean allows for very moist and warm maritime air getting lifted over the complex topography.

[revised manuscript text omitted]

troposphere levels in those regions, where high TD numbers are observed. Our results suggest that different large-scale conditions might favor convection in the various sub-regions. The substantial impact of the NAO has to be dicussed by considering dynamical and thermodynamical aspects separately. On the one hand, negative NAO phases are connected to flow patterns that are associated with frequent short-wave troughs providing lifting over Central Europe, whereas during positive phases a high geopotential anomaly tends to suppress convection due to a lack of large-scale lifting. On the other hand, negative phases go along with a pronounced reduction of $\theta_e$ at 850 hPa over most of the investigation area, which is detrimental to convective activity, while a positive $\theta_e$ anomaly is present during positive phases. Hence, the thermodynamical effects of the NAO phases on convective activity partly compensate for the dynamical effects. Owing to pronounced seasonal variations of the NAO pattern, the features discussed above are characteristic for the summer half-year only.

A potential weakness of our research is that the 14-year time series of lightning data does not reach the climatological time range. However, this sample should be sufficient for reliable analyzes regarding spatio-temporal variability on diurnal and seasonal time scales. As opposed to earlier studies, which all are based on shorter time spans, we were able to additionally investigate multi-annual modes of variability using well-suited significance tests in order to ensure that no misleading conclusions are drawn due to the limited sample size. Furthermore, the analyzes presented in this paper focus on CG lightning only. In fact, present-day LF location systems are capable of detecting IC lightning as well. For instance, Marra et al. (2017) studied IC activity within a strong convective cell using LF data. However, multi-annual statistical analyzes regarding IC lightning suffer from a significant reduction of both detection efficiency and location accuracy compared to CG lightning (Schulz et al., 2014). Although an IC climatology might potentially reflect further meteorological aspects, we therefore decided to neglect IC lightning. A further interesting issue is the spatio-temporal variability of winter lightning, particularly in view of the connections between thunderstorm genesis in marine regions and the seasonal variations of the temperature difference between water and air. However, much longer time series would be necessary for statistically reliable studies of winter lightning owing to the small event samples.

The clear and significant impact of the NAO pattern on convective activity in Europe motivates us to scrutinize the crucial role of large-scale flow in future research. The objective is to gain further insight into the physics behind the complex spatio-temporal variability discussed in this paper. This involves evaluating additional teleconnection modes obtained by Barnston and Livezey (1987) such as the East Atlantic Pattern. To generate longer, multi-decadal time series allowing for further investigations such as trend and spectral analysis, we have additionally implemented objective weather types as indicators for a high convective predisposition (Piper et al., 2016).

*Acknowledgements.* Funding by the Helmholtz Climate Initiative REKLIM (Regional Climate Change), a joint research project of the Helmholtz Association of German Research Centres (HGF) is gratefully acknowledged. We gratefully thank Siemens AG for providing lightning data and NOAA for providing NAO data. We acknowledge support from the Open Access Publishing Fund of KIT. We thank the two anonymous reviewers for their comments, which helped to improve the quality of the paper.

[revised manuscript text omitted]